# CoBot Studio VR: A Virtual Reality Game Environment for Transdisciplinary Research on Interpretability and Trust in Human-Robot Collaboration

### Martina Mara
LIT Robopsychology Lab
Johannes Kepler University
Linz, Austria
martina.mara@jku.at

### Kathrin Meyer
LIT Robopsychology Lab
Johannes Kepler University
Linz, Austria
kathrin.meyer@jku.at

### Michael Heiml
Polycular e.U.
Hallein,
Austria
michael.heiml@polycular.com

### Horst Pichler
Joanneum Research - Robotics
Klagenfurt, Austria
horst.pichler@joanneum.at

### Roland Haring
Ars Electronica Futurelab
Linz, Austria
roland.haring@ars.electronica.art

### Brigitte Krenn
Austrian Research Institute for
Artificial Intelligence, Vienna, AT
brigitte.krenn@ofai.at

### Stephanie Gross
Austrian Research Institute for
Artificial Intelligence, Vienna, AT
stephanie.gross@ofai.at

### Bernhard Reiterer
Joanneum Research - Robotics
Klagenfurt, Austria
bernhard.reiterer@joanneum.at

### Thomas Layer-Wagner
Polycular e.U.
Hallein, Austria
thomas.layer-wagner@polycular.com

## ABSTRACT
With the rise of collaborative robots (cobots) in industrial workplaces, designing these robots to meet the needs of their human co-workers is becoming increasingly important. Psychological theories and recent results from HRI research suggest interpretability and predictability of cobot behavior as key factors for the establishment of trust and collaborative task success. However, it is not yet clear which robotic intention signals are easily interpretable for whom and how they might impact user experience in varying situations. Therefore, the transdisciplinary team of the research project CoBot Studio created an innovative virtual reality (VR) environment in which the effectiveness of different light- and motion-based cobot signals can be systematically evaluated (i) in various collaborative mini-games, (ii) without safety concerns, (iii) under randomized controlled conditions, and (iiii) without the need to make actual hardware adjustments to a robot. This paper introduces functionalities and the system architecture of the highly versatile CoBot Studio research environment, presents an initial application, and discusses methodological benefits of using immersive VR games for research into human-robot collaboration.

## CCS CONCEPTS
• Human-centered computing → Human computer interaction → HCI design and evaluation methods • Human-centered computing → Visualization • Human-centered computing → Interaction design → Empirical studies in interaction design

## KEYWORDS
Human-robot collaboration, industry, virtual reality, interpretability, intention signaling, trust, multimodal communication, digital twin

**ACM Reference format:**

Martina Mara et al. 2021. CoBot Studio VR: A Virtual Reality Game Environment for Transdisciplinary Research on Interpretability and Trust in Human-Robot Collaboration. *VAM-HRI 2021, Boulder, Colorado USA*

## 1 Introduction

For the past half century, conventional industrial robots have been heavy expensive machines, usually kept away from human workers behind barriers or in cages. With the rise of collaborative robots (cobots), things are changing. Cobots are light and safe enough to work physically close with human co-workers, with each being able to use their specific skills on the same component at the same time in the same workspace. Typical applications of cobots include pick and place tasks, quality inspection tasks, packaging, welding or the assembly of objects together with a human partner. In contrast to a declining overall trend with traditional industrial robotics, the International Federation of Robotics reports a rapid

growth of cobot installations that has continued for several years [11]. However, although the field is expanding, collaborative robotics is still in its infancy with a market share of just 4.8% of total industrial robots installed in 2019 [11]. Thus, manufacturers, providers and users are still in need of gaining experience on what works in the implementation and design of collaborative robots and what doesn't.

## 1.1 Interpretability of collaborative robots

Now that the formerly isolated industrial robot is transforming into a close interaction partner for human workers, the establishment of mutual understanding is becoming critical. Collaboration between humans and robots can only be fluent, efficient and convenient if there is good communication and if both are able to comprehend each other's goals. That is, just as the states and intentions of the human partner must be identifiable for the robot, so too should the states and planned actions of the robot be easily understandable for the human co-worker [5][27][26]. A good interpretability of robot behavior also represents an important foundation for user acceptance and perceived trustworthiness of cobots. Being able to anticipate what a teammate is about to do and whether he or she (or it) will act in accordance with one's own expectations has been described as one of the key determinants of trust formation, both in psychological theories on cognition-based trust in interpersonal relationships [17][24] as well as in the recent Trust in Automation literature [13]. The increasing need for robots that are easily understandable in what they do and what they are about to do is putting more and more focus on the development and evaluation of robot intention signals.

Ideally, cobots must signalize their intentions in a manner that is also interpretable for people with limited experience. For instance, when a cobot is about to actively intervene in a work process, its actions, such as which direction it will move and which object it will grip, should be intuitively apparent to nearby persons [1][3][15][26]. Which robotic signals are readily interpretable by whom in which collaborative contexts still confronts researchers and industrial engineers with a need for greater insight. While there has been promising empirical work in recent years, for example on beneficial implications of light signals and projection-based intention signals [33], robot gestures [14], legible trajectory designs [5], or the use of augmented reality interfaces for the explanation of robot behavior [20], there is still a lack of systematic broad-scale studies assessing causal effects of various cobot intention signals under conditions in which all other factors—apart from the signals themselves—are held constant (e.g., type of robot, instruction, task, environment, light conditions).

Since the implementation of different intention signals in physical robots is often very complex and costly, and since field studies in real-life industry settings typically have to be rather condensed as otherwise the work flow might be disturbed, this goal is not easily achieved in the physical realm. The goal of the research project CoBot Studio therefore was to use highly immersive virtual reality (VR) to create a transdisciplinary research environment for the systematic investigation of parameters that contribute to good (hence interpretable, safe and trustworthy) human-robot collaboration.

## 1.2 Virtual Reality as a research tool

Virtual reality (VR) is described to hold immense promise for scientific research due to advantages offered in experimental control, reproducibility, and ecological validity [23][32]. Not least in the area of human-robot interaction (HRI), VR has emerged in recent years as an innovative research tool that complements empirical work in the field, online, or in the lab [6][10][18]. Especially in user studies with larger industrial robots or mobile manipulators, virtual environments constitute a safe space in which new signals and trajectory designs or higher intended robot speeds can be tested without safety concerns [22]. Recent improvements in the quality and accessibility of head-mounted displays and wide-area tracking systems make VR even more attractive for both research and real-world applications.

The ultimate aim of VR systems is to immerse the user into virtual worlds, inducing a sense of spatial presence, social presence and self-presence [7]. Numerous studies so far indicate that people develop a sense of "being there" in virtual environments [34], that they perceive virtual characters as actual social agents [9], and in some cases even show strong physiological reactions to stimuli presented in VR [16]. According to Slater and Wilbur [28], characteristics such as multisensory integration, interactivity, a wide visual field, high graphical resolution and color richness can additionally increase the immersive potential of VR. It is probably VR game developers in particular who regularly make use of these characteristics to create appealing products.

In order to create an immersive, realistic and enjoyable VR simulation environment, in which various experimental studies on industrial human-robot collaboration can be operationalized in the future, the transdisciplinary project team of CoBot Studio is composed not only of roboticists, psychologists, AI experts and multimodal communication researchers, but also of professionals experienced in VR programming, interaction design and game design. In the following, our research environment CoBot Studio VR is presented.

## 2 CoBot Studio VR

## 2.1 The idea

The fundamental idea behind the CoBot Studio project is to create an immersive research and simulation environment in which relevant target groups (e.g., industrial workers who will work with cobots in the future) meet virtual robots, perform tasks together with them, and simultaneously provide reports about their subjective experience of the respective robot. For this purpose, a high-resolution 360-degree interactive surrounding, modeled on an industrial working environment, was developed. Users should take part in collaborative mini-games, in which assignments such as guessing a cobot's intended target location or handing over objects have to be jointly executed as quickly and accurately as possible. During the game, nonverbal intention signals issued by a virtual cobot should be systematically manipulated in order to be able to assess their interpretability as well as their causal effect on task success, robot perception and user experience. To achieve a good

transferability of the evaluated cobot signals and psychological effects from VR to real-world contexts, virtual robots in the CoBot Studio game environment are implemented as digital twins of their physical counterparts. This approach allows on one hand for the simulation of real(izable) robot behavior in VR and on the other hand for the reuse of the robot control in physical realms.

## 2.2 Technical implementation

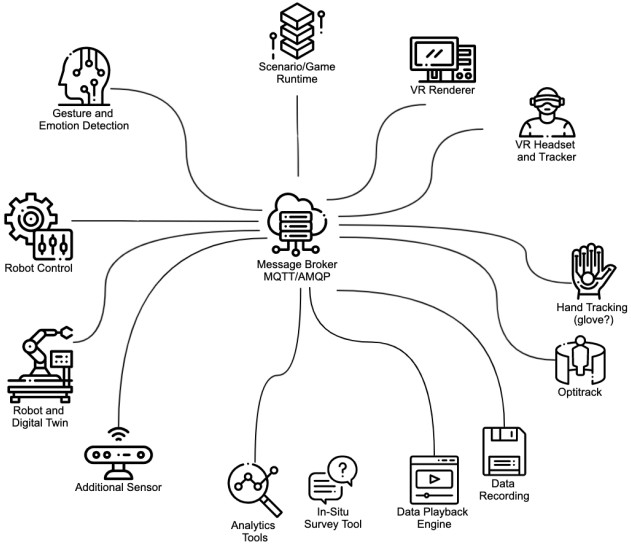

**Figure 1: Overview of components of the VR system.**

The CoBot Studio research environment demands a novel combination of hardware and software components (Figure 1). In consideration of future plans in this research project, the system architecture is designed to work for different modular software components and devices. Now used for a pure VR environment, it can be applied for later research with a physical robot and other tracking and interaction devices. All hardware and software endpoints communicate via the IoT network protocol AMQP [21]. All server software endpoints (e.g., robot controller, database, AMQP broker) are combined into a multi-container application by means of Docker Compose [4].

The VR environment was realized in the Unity game engine [31] and incorporates a lengthy list of features to support playfulness, immersion and the requirements of controlled experimental research on human-robot communication. To achieve a better gesture recognition than provided directly by the used head-mounted display (HMD) HTC VIVE Pro Eye, a Leap Motion infrared sensor was attached on the HMD front. This allows to use hand gestures in addition to the VR controller for interacting in the VR environment.

The robot's virtual digital twin is a 1:1 replica of the CHIMERA mobile manipulator, comprising a UR10 collaborative robot with a two-finger gripper, mounted on a MiR100 mobile platform, both with the same capabilities and looks as in real life. For the CoBot Studio VR environment robot models in URDF (Unified Robot Description Format) [30] for MiR100 and UR10 were integrated in a model for CHIMERA.

The goal of the chosen architecture was the development of a robot control that works for real robots in the same way as for their simulated counterparts (Figure 2; note that, for sake of brevity, the gripper is omitted as a third component besides UR and MiR). By using a unified ROS-based [25] messaging protocol, the simulated robots can be displayed in Unity via ROS# [2] and, if intended, in other commonly used visualization tools (such as RViz [8]) or simulation tools (such as Gazebo [12]). In the game configuration high-level robot actions can be combined into plan graphs for the required experiment tasks and variations, with parameters such as poses, durations and lengths inserted at runtime depending on the current target.

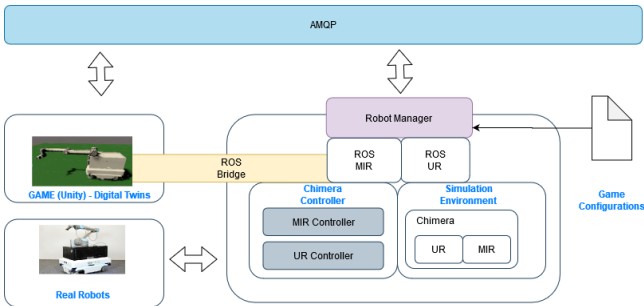

**Figure 2: Robot control integration in the system architecture.**

## 2.3 In-situ measurements and data collection

Various quantitative data can be collected during a CoBot Studio game run to allow for analysis according to the multiple involved scientific disciplines and their respective research questions and methodological approaches. The described technical realization intends to gather the recorded data of all measurement points in one central database to obtain all data linked to a specific game session.

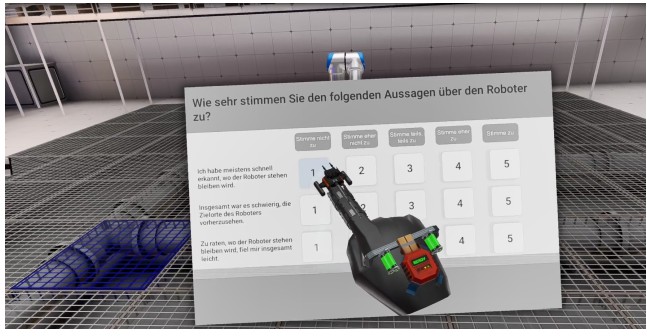

**Figure 3: Event-triggered in-situ survey in the VR environment.**

Information like sociodemographic variables, relevant personality traits, attitudes towards robots, and pre-experience with robots, games or VR are filled in by the participants themselves on

a tablet questionnaire before and after a game run. For all measures related to an individual's subjective experience of a cobot in the VR research environment, we implemented in-situ surveys. Survey items are triggered by specific game events (e.g., player has completed the first task of mini-game 3) and displayed live in scene at the appropriate moment. To answer the questions, hand tracking is used so that participants only need to click the virtual checkboxes with their fingers (Figure 3). Thereby, in-situ surveys are meant to catch the most direct user responses at the most relevant points in time.

Besides that, a VR game per se features several tracking and measurement options that are of use for the CoBot Studio research environment. For example, the integrated eye-tracking function of the HMD is used to control whether a participant is actually looking in the direction of the robot at the moment when the robot gives its intention signal. Results of this attention check can then be considered in conjunction with the participant's subjective robot evaluations and objective task performance. Based on the HMD and body trackers worn by the participant, the person's position can be determined. Combined with the known location of the virtual robot and other objects, distance measures are collected, e.g., the proximity between human and robot or the divergence of positions estimated by the human and targeted by the robot.

Events like the successful completion of a task or the recognition of a specific hand gesture are clearly defined in a VR game. This allows accurate measurement of time intervals, e.g. between a provided cobot signal and the person's reaction.

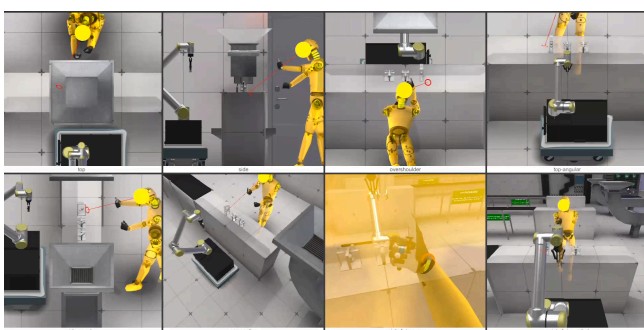

**Figure 4: Still of a multi-perspective video taken during a game run, showing the scene from different angles.**

To enable qualitative analysis of the human-robot interactions, a video of the VR environment including the robot and the human participant is recorded out of eight different perspectives (Figure 4). The VR environment combines several virtual camera feeds to record the experiment in a single video. A number of sensors allow tracking of the test participant. Five body trackers (wrists, waist, feet) plus the HMD are used for a virtual reconstruction of the body, including posture, head pose and arm gestures. For the representation of hand gestures, data from a Leap Motion sensor is used. Furthermore, by using the eye tracker built into the HMD, the human eye gaze is visualized in the video, providing insight into human gaze behavior. Altogether, the multi-perspective videos are a valuable source for the qualitative study of human nonverbal behavior in human-cobot interactions.

## 3    Initial application

With the described technical realization and measurement capabilities, CoBot Studio VR is set up as an environment in which various human-robot team tasks and different cobot intention signals can be evaluated in the context of mini-games. For a first user study, three interactive mini-games were conceptualized and implemented.

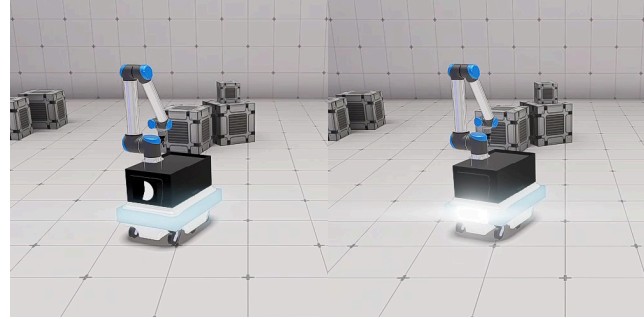

**Figure 5: Two examples of light signals designed to indicate a cobot's distance to its target location, with a circle that fills progressively on the left and a light spot, directed towards the target, becoming wider with decreasing distance, on the right.**

During these mini-games, the intention signals issued by the robots are varied, with each player being randomly assigned to one signal condition per mini-game, to evaluate their impact on human experience and collaborative task success. See Figure 5 for two examples of light signals for a mobile cobot.

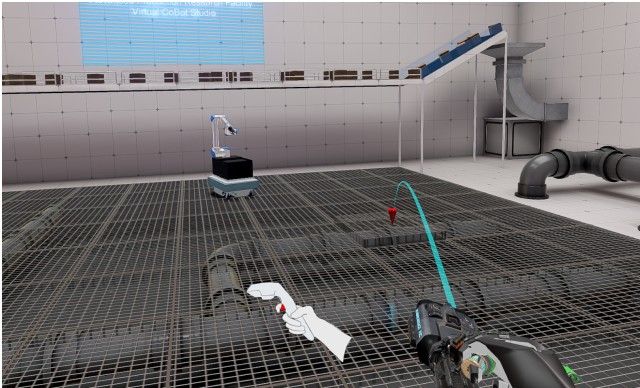

**Figure 6: A mobile cobot heads to a specific floor tile (player's perspective mini-game 1).**

Mini-Game 1 is called "Guess target location" (Figure 6) and features the following scenario: A mobile cobot platform moves towards a target floor tile in the room. As soon as the participant can guess where the robot heads to, he or she can stop the robot with a hand gesture and then point to the presumed targeted floor tile with the VR controller. Different light- and motion-based intention signals were designed, which the robot uses to proactively indicate its target. Depending on the different intention signals, group differences in the

objective and subjective predictability of the robot's goal and related effects on trust, subjective safety and user acceptance are explored.

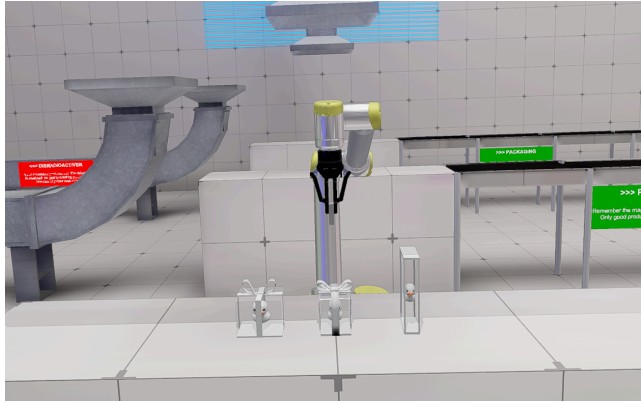

**Figure 7: A robot arm reaches for a specific object (player's perspective mini-game 2).**

In mini-game 2 ("Guess target object", Figure 7), a stationary robot arm reaches for one of several objects on a table in each of several rounds. The participant can stop the robot at any time via hand gesture and touch the object that he or she assumes is the one targeted by the robot. In this mini-game, different initial arm movements and gripper signals (e.g., the gripper adjusts itself to the affordance of the intended object's handle before the robot arm moves forward) were implemented as experimental signal conditions and again it is evaluated which leads to the highest interpretability and predictability of robot intentions.

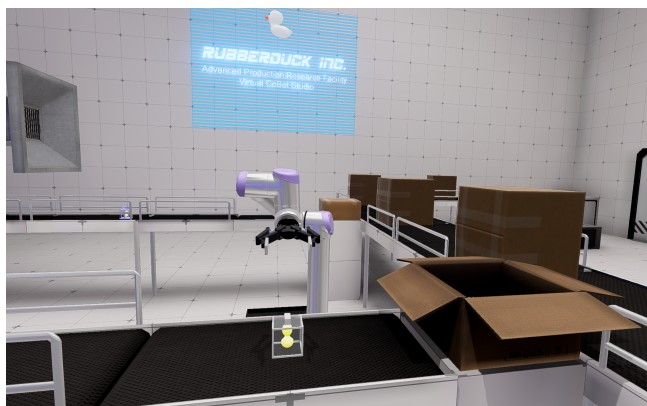

**Figure 8: The virtual cobot signalizes a player to take over (player's perspective mini-game 3).**

In mini-game 3 ("It's your turn", Figure 8), objects come along on a conveyor belt. A few times the virtual cobot picks them up and puts them in a cardboard box. After that, the robot tries to signalize that now it is the turn of the participant. Different motion- and light-based signals are experimentally varied in this mini-game to evaluate how well the participants understand the robot and how long it takes until they actually continue the task.

## 4  Discussion and outlook

With CoBot Studio, our goal was to create a novel research environment that unites high standards of professional VR game design with best practices of scientific method and combines hardware and software components in an innovative modular system architecture. Due to the flexibility of CoBot Studio, a broad variety of human-robot collaboration scenarios can be realized in the form of collaborative mini-games and various cobot intention signals can be tested easily by relevant user groups. The system permits the replacement of single components seperately, for example to add another set of cobot signals, to test an alternative VR headset in comparison, or to switch from a pure VR environment to mixed reality (MR) by replacing the virtual cobot representation with a physical one. Thereby, we have built a fundamental framework to support the study of human-robot collaboration in different levels of MR. This, in turn, could also add to existing research on the reliability and validity of VR/MR studies, when comparing empirical results of the same cobot intention signal evaluated in different levels of reality [19].

Using the approach of CoBot Studio, it is possible to build on a single control for the real cobot, its digital twin in a simulation or visualization, and combinations thereof. This allows us to let humans and robots work together virtually in playful environments where the absence of safety constraints gives the freedom to explore and compare novel manners of robot movement and communication between robot and human, and to easily transfer the results back into real-world scenarios, in which they can foster mutual understanding and also contribute to safety.

With regards to control and planning of robot motions, the three mini-games of our initial CoBot Studio application required us to overregulate aspects of robot movement that would be neglected in conventional systems: In some variations of mini-game 1 ("Guess target location"), the robot arm is used to point at a target location, which was realized based on the relative pose of the robot compared to the target while the mobile platform is moving. Mini-game 2 ("Guess target object") works with different designs of the arm and gripper movement that resulted from extracting cues from a motion and reassembling an overall motion with different patterns of temporal arrangement of the cues. E.g., the rotation of the whole arm or of the gripper would be done either separately early on or along with the forward movement of the end effector. In a similar fashion, the opening of the gripper from an initially closed position can be timed in various ways relative to the arm movement. Mini-game 3 ("It's your turn") uses a series of Cartesian end effector movements for pick-and-place actions, succeeded by manually designed signaling gestures. In certain conditions, the movements can be enriched with additional light signals in the Unity environment. The CoBot Studio VR environment therefore allows us to assess the interpretability of different cobot gestures, motion cues, or light signals without actual changes to the hardware and under highly controllable experimental conditions.

From the perspective of scientific data collection and data analysis, the integration of objective performance measures (e.g., how often or how fast a player guesses the correct target location of the robot in mini-game 1) and subjective self-reports (in-situ

questionnaires with standardized short scales, e.g., on trust, subjective safety or intention to use) as dependent variables can be considered particularly innovative. Instead of usual retrospective questionnaires, which in user studies often have to be answered some time after exposure to the experimental stimulus, the CoBot Studio in-situ surveys are displayed and answered at the relevant time triggered by a related game event in VR. Other advantages include automatic randomized assignment of players to one of several conditions per mini-game and consistent game instructions that take place directly in the VR environment and therefore minimize potential bias due to examiner effects.

The multi-perspective video recordings enable us to study non-verbal human behaviors in the respective collaborative situations, including postures and posture shifts, task-related human movements, gestures and gaze behaviors. Resulting findings will inform the definition and development of specific "robot behavior" components, which serve the control of (nonverbal) robot behavior in collaborative human-robot work situations, including the generation of robot-sided behavior (robot action and social signaling) and the robot-sided interpretation of collaboration-relevant nonverbal signals of humans in situations of task-oriented human-robot interaction.

Building on results on the interpretability of light- and motion-based cobot intention signals and their association with trust, subjective safety, and adoption intentions generated from experiments with the initial three mini-games, further iterations of the CoBot Studio research environment will be conceptualized and developed. Compared to our initial application, these should lead towards longer, more dynamic mini-games of human-robot collaboration as well as to translations of our CoBot games into different levels of reality along Milgram's reality-virtuality continuum [19]. Also, our robot control infrastructure, targeting physical robots as well as digital twins, shall more directly support the used types of goals and constraints for robot movements as inputs, e.g., requiring or forbidding certain axis movements within a certain specified time window of the overall movement. This way we can extend robot motion planning and control paradigms with new elements of the optimization problems that they incorporate.

Taken together, we think that CoBot Studio constitutes a highly flexible and so far unique research environment whose creation would not have been possible without the transdisciplinary composition of the project team. With practical guidelines yielded from our findings, we hope that our research can contribute to a human-centered design of cobots and collaborative work environments in the future.

## 5 Broader impact

The rising level of automation and digitization, sweeping through almost all areas of life, impacts the future of work. Robots, just like other machines and technological advancements, will constitute an even more integral part of many work environments than today. As close collaborations between humans and robots increase, the latter must be programmed to be easily understandable, predictable and thus trustworthy for people. To

make this a reality, transdisciplinary research is needed to develop empirically driven principles for user-friendly cobot design. With this vision in focus, CoBot Studio serves as an immersive virtual reality research environment that allows for playful simulation of collaborative scenarios with industrial cobots. It is well suited for user-centered evaluations of intention signals and the study of trust formation in human-robot teams. However, due to the highly versatile architecture of CoBot Studio, it is easily conceivable that also other topics of interest to the HRI research community (e.g., interaction with social robots, mind perception, proxemics) could be studied in a similar research environment.

## ACKNOWLEDGMENTS

We would like to thank our team members Tobias Hoffmann, Christine Busch, Julian Zauner, Matteo Lucchi, and Matthias Weyrer as well as former team members Robert Praxmarer, Christopher Lindinger, Franz Berger, Susanne Teufelauer, Peter Suchentrunk, Franziska Rabl, Ben Deetjen, Michael Kager, and Mathew Sherry for their contribution to CoBot Studio VR.
This work was supported by the IDEEN LAB program of the Austrian Research Promotion Agency (FFG).

Project information and videos: www.cobotstudio.at

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
