# OpenReview forum: "CoBot Studio VR: A Virtual Reality Game Environment for Transdisciplinary Research on Interpretability and Trust in Human-Robot Collaboration "
_humanrobotinteraction.org/HRI/2021I/Workshop/VAM-HRI — VAM-HRI 2021 Oral_

### Official Review · AnonReviewer1 · 2021-03-03
**CoBot Studio Review**

**Rating:** 9
**Confidence:** 5

**Review:**

This paper discusses CoBot Studio, a VR based human-robot interaction testbed/research environment that takes design from VR game design and modular system architecture. The paper described different aspects of CoBot Studio and their design for HRI. These include a variety of data collection modalities, in-situ measurements, and scenarios. Three collaborative human-robot tasks (“minigames”) are described within CoBot Studio (e.g., human identifying robot intent via different proposed non-verbal signals). Future outlook on CoBot Studio is also discussed highlighting the benefits of using CoBot Studio in future HRI research.

This paper is well written with a very extensive set of tech behind it. I think it could easily be used for extensive/repeatable HRI research. I don’t have a lot to add other than possibly adding a video link to the work within the paper (I know some people prefer not to). I would also love to see this open-sourced if at all possible.

Minor nitpicks below:

SIGVerse (https://arxiv.org/abs/2005.00825 ) is a highly applicable reference here and might help seat some of the work

“or the use of augmented reality interfaces for the explanation of robot behavior” -> should have some form of reference

References to ROS/Gazebo/ROS#(if used) etc should be included in my opinion

---

### Official Review · AnonReviewer2 · 2021-03-04
**Review: CoBot VR Environment**

**Rating:** 9
**Confidence:** 5

**Review:**

This paper presents a novel modular experimental VR environment where different robots, HRI techniques, interfaces, and platforms can be tested. It contains a thorough description of the environment, technical implementation, and data collection capabilities, as well as 3 different testbeds/games. It is well written and thorough. Check out the Situational Awareness Global Assessment Technique (SAGAT) developed by Endsley (1998) as a basis/reference for your in-situ surveys, as the technique you describe mirrors theirs. Is there a video that you can link to in the paper to demonstrate the full functionality? Strongly recommend accept.

---

### Decision · Program_Chairs · 2021-03-06

Accept (Oral)